# Development of a Point-of-Care Immunochromatographic Lateral Flow Strip Assay for the Detection of Nipah and Hendra Viruses

**DOI:** 10.3390/v17071021

**Published:** 2025-07-21

**Authors:** Jianjun Jia, Wenjun Zhu, Guodong Liu, Sandra Diederich, Bradley Pickering, Logan Banadyga, Ming Yang

**Affiliations:** 1National Microbiology Laboratory, Public Health Agency of Canada, Winnipeg, MB R3E 3R2, Canada; jianjun.jia@phac-aspc.gc.ca (J.J.); guodong.liu@phac-aspc.gc.ca (G.L.); logan.banadyga@phac-aspc.gc.ca (L.B.); 2National Centre for Foreign Animal Disease, Canadian Food Inspection Agency, Winnipeg, MB R3E 3M4, Canada; wenjun_zhu@hotmail.com (W.Z.); bradley.pickering@inspection.gc.ca (B.P.); 3Institute of Novel and Emerging Infectious Diseases, Friedrich-Loeffler-Institut, Federal Research Institute for Animal Health, Südufer 10, 17493 Greifswald, Germany; sandra.diederich@fli.de; 4Department of Medical Microbiology and Infectious Diseases, University of Manitoba, Winnipeg, MB R3E 0J9, Canada

**Keywords:** nipah virus, hendra virus, henipavirus, monoclonal antibody, ephrin B2, virus antigen detection, immunochromatographic lateral flow assay

## Abstract

Nipah virus (NiV) and Hendra virus (HeV), which both belong to the genus henipavirus, are zoonotic pathogens that cause severe systemic, neurological, and/or respiratory disease in humans and a variety of mammals. Therefore, monitoring viral prevalence in natural reservoirs and rapidly diagnosing cases of henipavirus infection are critical to limiting the spread of these viruses. Current laboratory methods for detecting NiV and HeV include virus isolation, reverse transcription quantitative real-time PCR (RT-qPCR), and antigen detection via an enzyme-linked immunosorbent assay (ELISA), all of which require highly trained personnel and specialized equipment. Here, we describe the development of a point-of-care customized immunochromatographic lateral flow (ILF) assay that uses recombinant human ephrin B2 as a capture ligand on the test line and a NiV-specific monoclonal antibody (mAb) on the conjugate pad to detect NiV and HeV. The ILF assay detects NiV and HeV with a diagnostic specificity of 94.4% and has no cross-reactivity with other viruses. This rapid test may be suitable for field testing and in countries with limited laboratory resources.

## 1. Introduction

*Henipavirus* is a genus within the *Paramyxoviridae* family that includes two zoonotic pathogens, Nipah (NiV) and Hendra virus (HeV). NiV and HeV are distinguished from other paramyxoviruses by their unique genetic constitution, biological features, high virulence, and broad host range. NiV and HeV infections can cause severe systemic, neurological, and/or respiratory diseases in humans and a variety of mammals, resulting in significant morbidity and mortality [1,2]. NiV is a potential pandemic threat due to its ability to spread from person to person, as well as its capacity for causing high mortality rates (70%) in humans [3]. HeV infection is a rare emerging zoonotic disease that can be transmitted from horses to humans, causing severe and often fatal disease in both [4]. Accordingly, NiV and HeV are classified as risk group 4 pathogens, restricting research to Containment Level 4 (CL4) laboratories [5,6]. In terms of prevention, there is currently no licensed vaccine for NiV infection; only a registered HeV vaccine for horses exists [7]. Therefore, due to the severity of NiV and HeV infections, the limitations of treatment, and the lack of available vaccines, surveillance and rapid identification of positive cases remain critical to preventing or limiting the impact of human outbreaks [8,9].

The diagnosis of henipavirus infection is made by monitoring clinical symptoms and laboratory testing. Currently, there are multiple laboratory-based methods for NiV and HeV detection, including virus isolation, reverse transcription quantitative real-time PCR (RT-qPCR), the antigen detection enzyme-linked immunosorbent assay (AgELISA), and early antibody detection. However, in countries where the disease is endemic, there are only a limited number of CL4 laboratories that can safely manage suspect samples [9]. In contrast, rapid point-of-care immunochromatographic lateral flow (ILF) tests are simple to use in the field, provide results within minutes, and eliminate the need for additional time to transport samples to a CL4 facility. During an outbreak, ILF testing can be a useful support tool to identify the virus as quickly as possible so that control measures can be implemented to minimize the spread of the virus [10].

The genomes of henipaviruses are non-segmented, single-stranded negative-sense RNA [11]. Henipaviruses contain six transcription gene units encoding six major structural proteins, namely the nucleocapsid protein (N), phosphoprotein (P), matrix protein (M), fusion protein (F), glycoprotein (G), and large protein or RNA polymerase (L). The P gene encodes three predicted non-structural proteins: C, V, and W [12,13]. Henipaviruses have two surface proteins that mediate cell entry: a tetrameric receptor-binding glycoprotein and a trimeric fusion protein. Ephrin B2, which binds to G, is the functional receptor for NiV and HeV [14,15]. Ephrin B2 has a high affinity for henipavirus G and has been shown to capture henipaviruses in immunoassays [15]. Recently, ephrin B2, along with a NiV-specific monoclonal antibody (mAb), was used to develop AgELISAs for the detection of NiV and HeV [16]. Our previous work also demonstrated that ephrin B2 could be used as a capture agent for NiV detection in an ILF test using a commercial generic rapid test device (Bioporto Diagnostics A/S, Hellerup, Denmark) [17]. Antigen-specific mAbs are widely used in the development of diagnostic tests; however, their production is time-consuming and labor-intensive. Compared with the production of mAbs, the use of recombinant proteins in diagnostic test development may make these assays relatively simpler [18]. In this study, we generated and optimized custom ILF test strips instead of using a commercial device to detect NiV and HeV. The customized ILF test was also partially validated.

## 2. Materials and Methods

### 2.1. Viruses and Preparation

NiV-Malaysia (-M, GenBank Accession No. AF212302), NiV-Bangladesh (B, GenBank Accession No. AY988601.1) and HeV (GenBank Accession No. NC_001906.3) were provided by the Centers for Disease Control and Prevention (CDC) in Atlanta, GA, USA, and the Public Health Agency of Canada (PHAC), Winnipeg, Canada Ebola virus (EBOV, variant Mayinga 76) and vesicular stomatitis virus (VSV) pseudotyped with the EBOV glycoprotein (VSV-EBOV GP) were provided by PHAC. VSV pseudotyped with NiV G and EBOV GP (VSV-NiV G/EBOV-GP) was provided by Reston Microbiology Laboratory, Reston, VA, USA. Recombinant Cedar virus (CedV) was kindly provided by Stefan Finke, Friedrich-Loeffler-Institute, Greifswald, Germany [19]. Inactivated and purified foot-and-mouth disease virus (FMDV) and avian influenza virus (AI) were provided by the National Centre for Foreign Animal Diseases (NCFAD), Canada. NiV, HeV and EBOV were inactivated using gamma irradiation (5M rad). AI viruses and FMDV were inactivated using binary ethyleneimine [20,21]. All work with infectious risk group 4 viruses was performed in the containment level 4 (CL4) laboratories of the Canadian Science Centre for Human and Animal Health, Winnipeg, Manitoba, Canada (CSCHAH), under the jurisdiction of Public Health Agency of Canada’s National Microbiology Laboratory (NML) and the Canadian Food Inspection Agency’s National Centre for Foreign Animal Disease (NCFAD). All work was performed according to institutionally approved standard operating protocols.

### 2.2. Recombinant Proteins

A soluble form of His-tagged recombinant human ephrin B2 was expressed and purified by GenScript Inc. (Piscataway, NJ, USA) [17]. Recombinant NiV G and M were produced as described elsewhere [17,20]. Recombinant NiV N and F were purchased from NativeAntigen, Kidlington, UK, and DAG-WT633, Creative Diagnostics, NY, USA, respectively.

### 2.3. Monoclonal Antibody Purification and Conjugation

Monoclonal antibodies were purified using a Protein-G affinity column (GE, Fairfield, CT, USA) and an AKIA chromatography system, as described previously [21]. The purified mAbs were then conjugated to colloidal gold using a high-sensitivity conjugation kit for lateral flow (nanoComposix, Inc., San Diego, CA, USA) [22].

### 2.4. Lateral Flow Strip Preparation and Test Procedure

Lateral flow immunochromatographic assay test strips are composed of a backing card, a nitrocellulose membrane (NC, Whatman FF170HP, LFM-004, Cytiva, Marlborough, MA, USA), a sample pad (grade 8980, Ahlstrom, Helsinki, Finland), an absorption pad (whatman GF/DVA, Cat#8145-2250, Cytiva, Marlborough, MA, USA), and a conjugate pad (grade 6614). Test and control lines were sprayed to the nitrocellulose (NC) membrane with recombinant ephrin B2 (10 µg/cm) and the mAb against mouse IgG (0.5 µg/cm, Sigma-Aldrich, St. Louis, MO, USA), using a CAMAG Linomat 5 dispenser (CAMAG Scientific Inc., Wilmington, NC, USA). The conjugate pad was pretreated with 5% sucrose, 0.5% BSA, and 0.1% Tween 20 in PBS (pH7.4), and dried. Then, the conjugate pad was sprayed with colloidal gold-conjugated NiV G detection-specific mAb (F20NiV-65), and then dried for 2–4 h at 30 °C in a convection oven and stored at room temperature under dry conditions. All the components were assembled, cut into 4.0 mm width sections and kept in a desiccator at room temperature for further use.

For the assay, each sample (40 μL) was mixed with an equal volume of a BSA-based sample running buffer for the lateral flow assay (Bioassay Works LLC, Ijamsville, MD, USA) in a test tube, and then the sample was loaded into an ILF cassette, or the test strip was dipped into the test tube. The results were determined through visualization after 10–15 min. A positive result was indicated by bands on both the test and control lines. A negative result was indicated by a single band on the control line only. Results are interpreted as +/− (possible lines), + (visible lines), ++ (clearly visible lines), and +++ (strongly visible lines).

### 2.5. Enzyme-Linked Immunosorbent Assay (ELISA)

The indirect ELISAs and the NiV and HeV antigen detection ELISA were performed as previously described [16].

### 2.6. Negative Samples

Negative control (uninfected) oral/nasal swab samples (80 from pigs, 36 from guinea pigs, and 11 from ferrets) were previously collected at the National Centre for Foreign Animal Disease (NCFAD) in Canada. In total, 50 specific pathogen-free porcine sera collected at NCFAD were obtained from a farm in Winnipeg, MB, Canada.

## 3. Results

### 3.1. Characterization of NiV-Specific Monoclonal Antibodies

Seven NiV-specific mAbs were previously generated and selected for further evaluation. Five of the seven mAbs reacted with NiV (-B. -M) only, while the other two mAbs became bound to both NiV and HeV in an indirect ELISA. The target antigens of five (F20NiV-65, F27NiV-34, F45G-4, F45G-5, and F45G-6) of the seven mAbs had been previously identified using indirect ELISA or Western blot analysis (Table 1). They react with NiV structural proteins G, F, N, and M, respectively [16,17,20]. The binding sites of F20NiV-6 and F20NiV-80 were determined by ELISA against the NiV recombinant structural proteins G, N, and F. The results showed that the two mAbs bind specifically to NiV F but not to other structural proteins (Table 1).

### 3.2. Development of ILF Assay

Recombinant ephrin B2 was used as a capture agent and sprayed on the NC membrane test line. To determine which mAb could be used as the detection mAb, seven NiV-specific mAbs were purified and conjugated with colloidal gold. The testing result showed that among the seven mAbs, only F20NiV-65 [17] showed a clearly visible band on the test line of the strip. All other mAbs showed no signal or a very weak signal. Therefore, F20NiV-65 was chosen as the detection agent.

The strip assembly is shown in Figure 1. To find the optimal concentration of ephrin B2 on the control line, different concentrations of ephrin B2 in the range of 1.5–10 µg/cm were tested. The strongest reaction without a false-positive result was seen with ephrin B2 at a 10 µg/cm concentration on the test line. To optimize the detection agent, F20NiV-65 was conjugated with different sizes of colloidal gold particles. Various combinations of 80 nm and 150 nm gold particle-conjugated mAb were applied to the conjugate pads and tested. The best results were achieved with a 2:1 ratio of 80 nm to 150 nm gold-conjugated mAb.

### 3.3. Specificity of the ILF Assay

The ILF assay was evaluated using inactivated viruses (NiV-B, -M and HeV) and VSV-NiV G/EBOV-GP, all of which produced positive results (Figure 2), confirming that the assay can specifically detect NiV and HeV or their glycoproteins. Conversely, EBOV, VSV-EBOV-GP, CedV, AI H5N2, AI H7N3, and FMDV A_22_ all tested negative (Figure 2). Porcine serum and human plasma were spiked with NiV-M, NiV-B, and HeV and tested using the ILF assay (Figure 3). All virus-spiked samples were positive, with a similar band intensity to that of viruses diluted in PBS (Figure 3a). This suggests that the ILF assay may be able to detect NiV and HeV in serum and plasma.

Next, a total of 127 swab samples from uninfected animals, along with 50 negative pig sera, were tested to determine the diagnostic specificity (DSp) of the ILF test (Table 2). Of the 127 swab samples, 117 were negative and 10 were false positives (with very faint signals on the test line). All pig sera were negative. The DSp for the ILF test was therefore calculated to be 92.1%.

### 3.4. Analytical Sensitivity of the ILF Test

To determine the limit of detection (LOD), inactivated virus stocks (NiV-M, NiV-B, and HeV) were two-fold serially diluted and then tested using the ILF assay. The results showed that the test line intensity was dose-dependent, with the band at the test line ranging in intensity from strong to weak to negative for NiV and HeV (Figure 4). The LOD for NiV-M and NiV-B was 3.71 and that for HeV was 4.44 Log10 plaque forming units (pfu)/mL. By comparison, the LOD of our previously developed AgELISA [16] was 3.10–4.14 log10 pfu/1 mL (Figure 4). Thus, overall, the analytical sensitivity of the ILF test was not significantly different from that of the AgELISA.

## 4. Discussion

This report describes the development of an ILF assay using ephrin B2 and a NiV-specific mAb for rapid NiV and HeV detection. Rather than using a commercially available generic lateral flow device, we created a custom test strip, spraying ephrin B2 onto the test line and applying the colloidal gold-conjugated mAb onto the conjugation pad. Users can directly apply the sample onto the test device and obtain results within minutes.

To select the best mAb for the ILF assay, a panel of mAbs previously generated from mice immunized with inactivated NiV-M was further characterized [17,20]. All seven mAbs were colloidal gold-conjugated and tested using the ILF test strip with ephrin B2 on the test line. The results showed that only one of the seven mAbs could detect NiV when paired with Ephrin B2. Therefore, the mAb (F20NiV-65) that specifically binds to NiV G was selected as the detection agent for the ILF test. It is known that NiV structural protein N is the most abundant protein produced during NiV infection and is one of the most highly immunogenic NiV proteins [23,24]. However, the two NiV N-specific mAbs failed to detect NiV and HeV when incorporated in the ILF assay. The possible reason is that the NiV nucleocapsid, which contains viral RNA and associated proteins (N, P, and L), is surrounded by a lipid bilayer membrane (virus envelope), and therefore the virus may need to be lysed before testing. Although several mAbs recognized NiV in ELISA, only F20NiV-65 recognized NiV in the ILF format. The inability of the other antibodies to function in the ILF test may be due to the relatively brief incubation time between the antibody and antigen compared to the much longer incubation time in the ELISA. Due to the extremely short contact time, the antibody binding kinetics in the ILF assay have a significant impact on the assay results. In addition, the antibody must remain active after binding to the colloidal gold particles, maintain structural integrity when dried, and react immediately after rehydration in the ILF assay [10].

The customized ILF test, which relies on F20NiV-65 as the detection antibody, specifically detected NiV (-M, -B), HeV, and VSV pseudotyped with NiV G/EBOV GP. However, in the context of our previously published NiV-Ag ELISA, mAb F20NiV-65 became bound only to NiV but not to HeV when paired with Ephrin B2 [16]. The reason why F20NiV-65 showed different specificities in different assays may be due to the alteration or disruption of antibody binding under each assay condition. Factors such as pH, temperature, salt concentration, and the presence of detergents can all affect the strength and specificity of the antibody–antigen interaction. Specifically in this case, the altered binding profile of F20NiV-65 could be because of (1) the detergent used in the AgELISA, as detergents disrupt hydrophobic interactions and, in some cases, reduce the effectiveness of antibody binding; (2) the plastic wells used in the AgELISA, since binding of viruses to the well surface may change the conformation of the protein and affect antibody binding; or (3) conformational changes in F20NiV-65 following binding to colloidal gold particles, which can affect the strength and specificity of the interaction with the antigen.

Notably, the ILF assay is specific to NiV and HeV, without cross-reaction with other viruses, particularly CedV. CedV is a non-pathogenic henipavirus that was isolated in Australia in 2012. CedV is closely related to HeV and NiV, with approximately 48% nucleotide sequence homology and approximately 30% amino acid identity to the NiV/HeV glycoprotein [25,26]. Despite the homology between CedV and NiV and HeV, the current ILF assay does not cross-detect CedV, even with ephrin B2 as the capture agent. Unfortunately, we were unable to test for cross-reactivity with other closely related henipaviruses, such as Langya virus (LayV) and Mojiang virus (MojV), because we do not have access to these viruses. While the inability of the ILF test to detect EBOV, FMDV, and AIV does not necessarily suggest differential diagnostic potential, it does further demonstrate the specificity of our ILF test for NiV and HeV.

NiV can be transmitted through direct contact with infected animals or through body fluids (blood, urine, or saliva) [27]. Thus, it is crucial that a diagnostic test can detect virus in those samples. During acute infection, NiV can be detected in blood samples, nasal swabs, cerebrospinal fluid, and urine by RT-qPCR [9]. Therefore, we also sought to determine if the ILF could detect NiV and HeV in serum or plasma samples. Due to the lack of serum samples from infected animals, we spiked porcine serum and human plasma with NiV and HeV. The results indicated that the ILF assay was able to detect NiV/HeV with no reduction in signal compared to that of the virus in PBS, suggesting that the ILF test may be able to detect NiV in serum or plasma.

Overall, the diagnostic specificity of the ILF test reached 94.4% based on the negative swab and serum samples. No false positives were detected in serum samples, probably because these samples are cleaner than other swabs (oral, nasal, and rectal swabs), which often contain various microorganisms, especially bacteria or other substances. In fact, 7.87% (10/127) of the swab samples produced false positives (Table 2). Excluding serum samples, the specificity of swab samples was 92.1%. The specificity of other sample types, such as cerebrospinal fluid (CSF), nasopharyngeal aspirates, or bronchial aspirates, remains unclear and needs further evaluation. Furthermore, due to the limitations of obtaining positive swabs and serum samples from animal experiments or epidemics, we were unable to fully validate our ILF assay. Additional testing using true-positive samples is needed to confirm that the ILF assay can detect NiV and HeV in field samples.

The main advantage of ILF testing compared to other viral detection technologies, such as virus isolation and ELISA, is that ILF assays are rapid and easy to run, providing results within minutes without the use of specialized equipment. In terms of point-of-care testing for NiV and HeV detection, several rapid molecular diagnostic tests based on reverse transcriptase recombinase isothermal amplification combined with lateral flow have been recently developed [28,29]. However, the molecular tests deployed in the field still require special equipment and training. We developed a customized ILF assay rather than using a generic, commercially available ILF device. This approach is less expensive and allows us to respond faster to diagnostic needs without having to rely on external vendors. Most importantly, users can directly apply the sample in running buffer to the test strip, rather than mixing the sample and reagents in a tube and then applying the mixture to the device.

The current ILF assay could not differentiate between NiV and HeV. This is because our ILF assay targets the NiV G gene, which is highly conserved between the two viruses, with an amino acid similarity of 79% [30]. Although it would be ideal to have a point-of-care ILF test that can specifically identify NiV or HeV in the outbreak setting, the present ILF test would still be able to quickly detect either virus, providing potentially critical information in support of outbreak control. Future work will focus on the development of mAbs specific for either NiV or HeV.

This rapid ILF test can quickly identify NiV and HeV. We anticipate that this test can be performed in the field, and field-available ILF tests do not require a cold chain, making them ideal for remote areas. During an outbreak, the ILF test can be a useful support tool to identify the virus as quickly as possible so that control measures can be implemented as quickly as possible to minimize the spread of the virus.

## Figures and Tables

**Figure 1 viruses-17-01021-f001:**
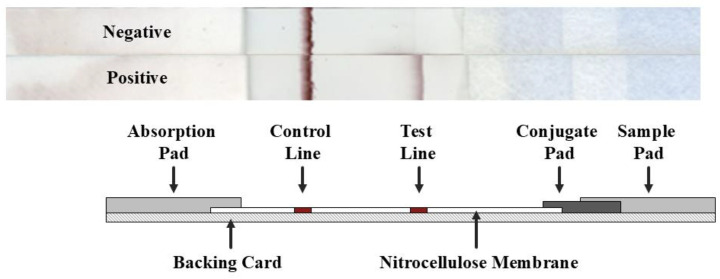
Schematic diagram of the ILF assay for the detection of NiV/HeV. When a sample containing the target virus is applied onto the sample pad, the colloidal gold-conjugated detection antibody in the conjugate pad binds to the virus, which then passes through the nitrocellulose membrane by capillary action. The target virus is captured on the test line (T) along with the colloidal gold-conjugated antibody and forms a visible band on the test line. The signal in the control line (C) comes from the same gold-conjugated antibody but does not contain the target analyte.

**Figure 2 viruses-17-01021-f002:**
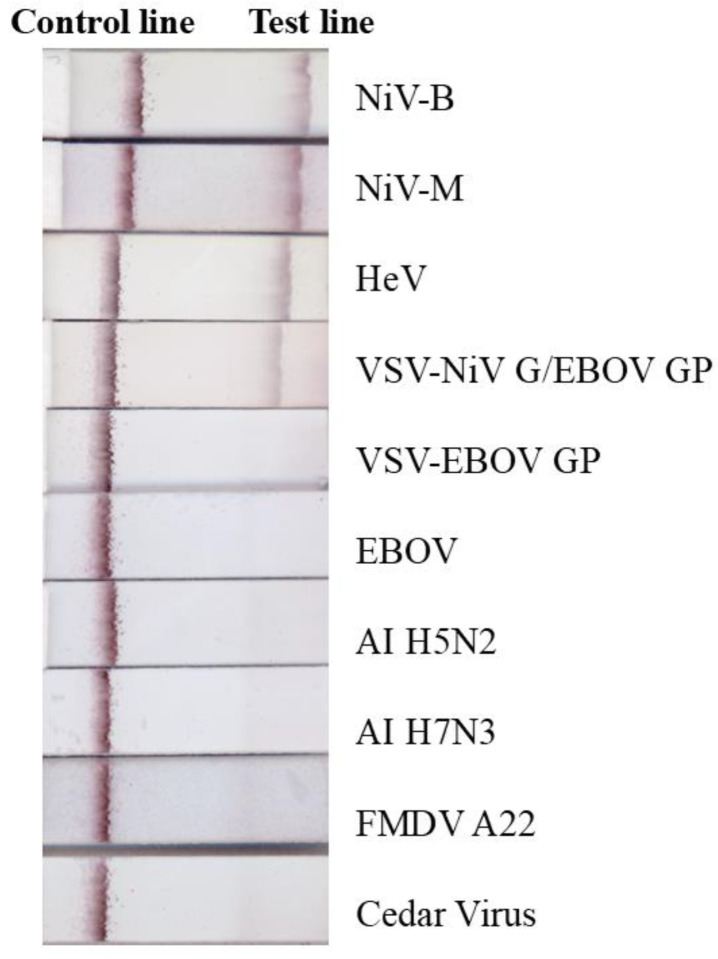
Specificity of the ILF assay. Inactivated NiV-B (4.61 log10 pfu), NiV-M (4.62 log10 pfu), HeV (5.35 log10 pfu), VSV-NiV-G/EBOV-GP (5 log10 pfu), VSV-EBOV-GP (3.3 log10 pfu), EBOV (5.18 log10 50% tissue culture infectious dose), and Cedar virus (5.7 log10 pfu), as well as purified and concentrated avian influenza virus (AI) H5N2 (1 µg), H7N3 (1 µg), foot and mouth disease virus (FMDV A22, 1 µg), were mixed with running buffer and loaded onto the ILF device or strip, and the results were read after 15 min.

**Figure 3 viruses-17-01021-f003:**
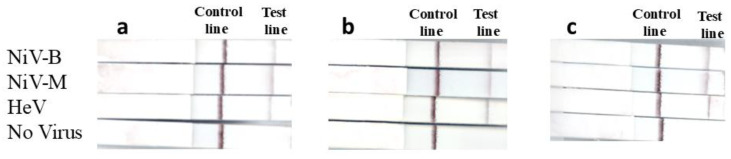
ILF detection of serum/plasma spiked with NiV and HeV. Viruses were spiked into (**a**) PBS, (**b**) negative porcine serum, and (**c**) normal human plasma, and the samples were tested using the ILF assay, with results determined after 15 min.

**Figure 4 viruses-17-01021-f004:**
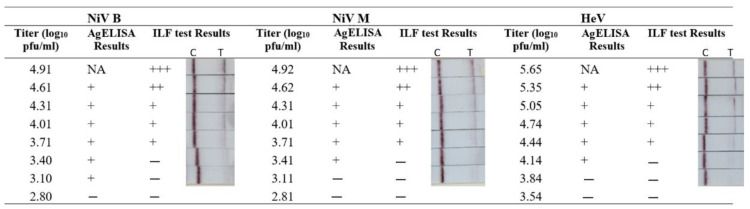
Comparison of analytical sensitivity of the ILF assay and antigen detection ELISA (AgELISA). Inactivated NiV and HeV were two-fold serially diluted in PBS. Each sample was tested in parallel using the ILF assay and AgELISA. In the AgELISA, optical densities ≥0.12 and ≥0.07 were considered positive results (+) for NiV and HeV, respectively. For the ILF assay, two visible bands on both the test line (T) and the control line (C) indicated a positive result. The C line is visible, indicating a negative result. (+++, strongly positive; ++, moderately positive; +, weakly positive; −, negative result).

**Table 1 viruses-17-01021-t001:** Reactivity of monoclonal antibodies against inactivated NiV/HeV and recombinant NiV structural proteins.

mAb	Isotype	Target Protein	NiV-M	NiV-B	HeV	G	N	F	Reference
F20NiV-6	IgG2a/ƙ	F	+	+	−	−	−	+	This study
F20NiV-65	IgG2b/ƙ	G	+	+	−	+	−	−	Yang et al. [17]
F20NiV-80	IgG1/ƙ	F	+	+	−	−	−	+	This study
F27NiV-34	IgG1/ƙ	F	+	+	+	−	−	+	Zhu et al. [16]
F45G-4	IgG1/ƙ	N	+	+	−	−	+	−	Berhane et al. [20]
F45G-5	IgG2a/ƙ	M	+	+	−	−	−	−	Berhane et al. [20]
F45G-6	IgG2a/ƙ	N	+	+	+	−	+	−	Berhane et al. [20]

**Table 2 viruses-17-01021-t002:** Negative sample list and ILF assay results.

Animal Species	Sample Source	Sample Number	ILF Test Results	
			Negative	False Positive
Ferret	Swab (oral, nasal, rectal)	11	9	2
Guinea pig	Swab (oral, nasal)	36	33	3
Pig	Swab (oral)	40	37	3
Pig	Swab (nasal)	40	38	2
Pig	Serum	50	50	0
Total		177	167	10

## Data Availability

The data sets generated and/or analyzed in this study will be made available upon reasonable request. A material transfer agreement may be required.

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
