# Peer review of "Development of a Point-of-Care Immunochromatographic Lateral Flow Strip Assay for the Detection of Nipah and Hendra Viruses"

_viruses, 2025, doi:10.3390/v17071021_

Round 1
Reviewer 1 Report (Previous Reviewer 2)
Comments and Suggestions for Authors
The article is an expanded version of the previous submission on the development of a diagnostic method for Nipah and Hendra viruses based on ILF.
In this version, the authors supplemented the article with a discussion of the reasons for the different specificity of F20NiV-65 antibodies and described the reason for choosing the viruses they used to test specificity.
In general, in the answers and additions to the text of the article, the authors answered the questions I raised in the previous version in the "General concept comments" section.
However, the questions from the "Specific comments" section remained unanswered and unchanged in the text, so I repeat them again
Line 205 where from is 124 swab samples? In methods 80 from pigs + 36 from guinea pigs + 11 from ferrets) = 127
When you are writing about LOD it is better to give ref some other methods of detection LOD. Compare results not only with yours ELISA, but with other articles or commercial kits. So that the reader understands whether you have achieved a good result
It is unclear what for the section "Characterization of NiV-specific monoclonal antibodies" is present in the article. It briefly describes the specificity of two antibodies F20NiV-6 and F20NiV-80, which are not subsequently used in the development.
If this is an article about the development of a test system and as a result was selected an antibody F20NiV-65 that was already described in the authors' previous article, you can write in the methods that such an antibody was used and a add a ref.
Author Response
We would like to thank reviewers for taking the time to carefully read our manuscript. We have revised the manuscript based on the comments of the reviewers. We appreciate the opportunity to resubmit the revised manuscript now.
Question 1
However, the questions from the "Specific comments" section remained unanswered and unchanged in the text, so I repeat them again
Line 205 where from is 124 swab samples? In methods 80 from pigs + 36 from guinea pigs + 11 from ferrets) = 127
Answer:
The total number of swab samples was changed from 124 to 127.
Question 2
When you are writing about LOD it is better to give ref some other methods of detection LOD. Compare results not only with yours ELISA, but with other articles or commercial kits. So that the reader understands whether you have achieved a good result
Answer:
Thank you for your valuable suggestion. This is a great idea, and we completely agree that comparing LOD results with commercial or other methods is the best comparison method, rather than just comparing with our own ELISA method.
However, at that time, we could not find any commercial ELISA kits that could detect Nipah virus, so we could not compare our results. Recently, we found a commercial ELISA kit for porcine Nipah virus (NiV). In the future, we will always consider using commercial kits to compare with any of our internal test results to verify that our detection method performs as well as expected.
Question 3
It is unclear what for the section "Characterization of NiV-specific monoclonal antibodies" is present in the article. It briefly describes the specificity of two antibodies F20NiV-6 and F20NiV-80, which are not subsequently used in the development.
If this is an article about the development of a test system and as a result was selected an antibody F20NiV-65 that was already described in the authors' previous article, you can write in the methods that such an antibody was used and a add a ref.
Answer:
We agree. Yes, the characterization of mAb F20NiV-65 has been published previously. The reference for this mAb has been added. This reference is also listed in Table 1.
Reviewer 2 Report (Previous Reviewer 3)
Comments and Suggestions for Authors
The authors have thoroughly addressed all the reviewers' comments in the revised manuscript. Therefore, I have no further remarks.
Author Response
We would like to thank reviewers for taking the time to carefully read our manuscript. We have revised the manuscript based on the comments of the reviewers. We appreciate the opportunity to resubmit the revised manuscript.
This manuscript is a resubmission of an earlier submission. The following is a list of the peer review reports and author responses from that submission.
Round 1
Reviewer 1 Report
Comments and Suggestions for Authors
The manuscript focuses on developing a rapid test for the detection of Henipavirus infection. The experiments appear well conducted, and the results are well presented. However, the paper lacks novelty, as similar work has already been published (e.g., https://pmc.ncbi.nlm.nih.gov/articles/PMC9949527/).
What distinguishes this assay from previous ones? Is it an improvement over the existing rapid tests?
The English in the manuscript is clear and well-structured. However, there are a few minor grammar, clarity, and formatting issues that could be improved for better readability and precision. Please revise.
Author Response
We would like to thank reviewers for taking the time to carefully read our manuscript. We have significantly revised the manuscript based on the comments of the reviewers. We appreciate the opportunity to resubmit the revised manuscript now.
Question: What distinguishes this assay from previous ones? Is it an improvement over the existing rapid tests?
Answer:
In our previous paper, our goal was to approve the concept that the receptor Ephrin B2 could be used as a capture agent for NiV viral antigen detection in immunoassays. In the previous report, we used a generic commercial device (RapidAssay device purchased from Rapid Assays, Copenhagen, Denmark) for NiV detection. In the generic device, the test line contained a biotin-binding protein that captured a biotinylated capture monoclonal antibody. The control line contained anti-mouse IgG antibody.
In the current report, we described the customized ILF assay for rapid NiV and HeV detection using Ephrin B2 and a NiV-specific mAb. Instead of using a commercially available generic lateral flow device, we created a custom test strip where Ephrin B2 is sprayed onto the test line, and the colloidal gold-conjugated mAb is applied to the conjugate pad. The user can directly apply the sample to the test device and obtain the results within minutes.
This approach is less expensive and allows us to respond faster to diagnostic needs without having to rely on external suppliers. Most importantly, users can add the sample and buffer directly to the test strip, rather than mixing the sample and reagents in a tube and then adding them to the device.
Question: The English in the manuscript is clear and well-structured. However, there are a few minor grammar, clarity, and formatting issues that could be improved for better readability and precision. Please revise.
Answer:
Some grammar, clarity, and formatting issues have been corrected.
Reviewer 2 Report
Comments and Suggestions for Authors
A brief summary
The article describes the development of a diagnostic method for Nipah and Hendra viruses based on ILF. The authors describe the verification of antibodies obtained earlier during the development of a system for the diagnosis of these same viruses using ELISA. For validation, the authors use swabs and serum from healthy animals with the introduced viruses inactivated using gamma irradiation, or VSV or a recombinant virus. The system has been shown to be specific to heterologous viruses. Such ILF system can be used for diagnostics because it is convenient to use in the field in conditions of insufficient equipment.
However, to improve the work, it would be good for the authors to make a number of adjustments and answer a number of questions that readers may have.
General concept comments
Nipah virus is found in Bangladesh, India, Malaysia, Philippines, and Singapore. Hendra virus is found only in Australia (Queensland and New South Wales states).
Is differential diagnostics carried out in these regions? Or is it enough for them to know whether it is Nipah or Hendra for diagnostics? I think that it is important for them what sp of virus they found. It is logistically unprofitable to check it later with the ELISA method, you can then look at it right away. Especially since the ELISA you developed forces the user to also run two tests in parallel to differentiate the viruses.
It would be much more logical to make antibodies specific to Nipah virus and separately antibodies specific to Hendra. Such a system would be much more relevant for use
It is not entirely clear for what purposes this diagnostic is intended. And where? In Canada? Is it possible to order a ready-made product in the form of a test system? Can researchers from Bangladesh, India, Malaysia, Philippines Singapore Australia order all the necessary antibodies, recombinant proteins and other consumables to recreate the diagnosticum locally?
How can you explain that F20NiV-65 antibodies, for which specificity only to Nipah virus was proven in your previous article, become non-species specific under these conditions and detect both Nipah and Hendra? If it is shown that they are non-specific, perhaps it is worth making corrections to the previous article (Zhu, W.; Smith, G.; Pickering, B.; Banadyga, L.; Yang, M. Enzyme-Linked Immunosorbent Assay Using Henipavirus-Receptor EphrinB2 and Monoclonal Antibodies for Detecting Nipah and Hendra Viruses. Viruses 2024, 16, 794. https://doi.org/10.3390/v16050794), that is, NiV AgELISA can detect not only NiV. If they are not species specific, it can be assumed that they will be able to detect similar viruses.
Thusit may be necessary to check for cross-reactivity with other viruses besides Cedar virus (CedV), for example such as Parahenipaviruses Langya virus and Mòjiāng virus, which are found in China.
What is the rationale for choosing EBOV for specificity comparison? Or foot-and-mouth disease virus (FMDV) and avian influenza virus (AI)? Are they somehow included in the list of potential differential diagnostics? It would be good to explain the choice of these viruses
Specific comments
Line 201 where from is 124 swab samples? In methods 80 from pigs + 36 from guinea pigs + 11 from ferrets) = 127
When you are writing about LOD it is better to give ref some other methods of detection LOD? Compare results not only with yours ELISA, but with other articles or commercial kits. So that the reader understands whether you have achieved a good result
It is unclear what for the section "Characterization of NiV-specific monoclonal antibodies" is present in the article. It briefly describes the specificity of two antibodies F20NiV-6 and F20NiV-80, which are not subsequently used in the development.
If this is an article about the development of a test system and as a result was selected an antibody F20NiV-65 that was already described in the authors' previous article, you can write in the methods that such an antibody was used and a add a ref.
Author Response
We would like to thank reviewers for taking the time to carefully read our manuscript. We have significantly revised the manuscript based on the comments of the reviewers. We appreciate the opportunity to resubmit the revised manuscript now.
Question: Nipah virus is found in Bangladesh, India, Malaysia, Philippines, and Singapore. Hendra virus is found only in Australia (Queensland and New South Wales states). Is differential diagnostics carried out in these regions? Or is it enough for them to know whether it is Nipah or Hendra for diagnostics? I think that it is important for them what sp of virus they found. It is logistically unprofitable to check it later with the ELISA method, you can then look at it right away. Especially since the ELISA you developed forces the user to also run two tests in parallel to differentiate the viruses.
Answer:
We agree with the reviewer. While it is critical that a point-of-care ILF test can specifically identify Nipah virus (NiV) or hepatitis E virus (HeV) in an outbreak, it would make more sense to produce monoclonal antibodies against Nipah and Hendra viruses. Such a system would be more suitable for rapid diagnosis. But at least the current ILF test can quickly detect whether the outbreak virus is one of them (NiV/HeV). Eventually, we will produce monoclonal antibodies against Nipah and Hendra viruses. This will allow us to develop new NiV/HeV ILF tests that are more convenient to use in the field.
Question: It is not entirely clear for what purposes this diagnostic is intended. And where? In Canada? Is it possible to order a ready-made product in the form of a test system? Can researchers from Bangladesh, India, Malaysia, Philippines Singapore Australia order all the necessary antibodies, recombinant proteins and other consumables to recreate the diagnosticum locally?
Answer:
The assay has the benefits of a rapid time to result and ease of use. In an outbreak of any magnitude, a field-based rapid diagnostic assay would allow proper patient transport and for safe burials to be conducted without the delay caused by transport of samples between remote villages and testing facilities. The use of such point-of-care instruments in South Asia/Australia would have distinct advantages in the control and prevention of local outbreaks.
Currently, recombinant Ephrin B2 and a panel of NiV-specific monoclonal antibodies have been prepared for use in NiV and HeV diagnostic applications. We can provide these reagents regardless of the customer's needs.
Question: How can you explain that F20NiV-65 antibodies, for which specificity only to Nipah virus was proven in your previous article, become non-species specific under these conditions and detect both Nipah and Hendra? If it is shown that they are non-specific, perhaps it is worth making corrections to the previous article (Zhu, W.; Smith, G.; Pickering, B.; Banadyga, L.; Yang, M. Enzyme-Linked Immunosorbent Assay Using Henipavirus-Receptor EphrinB2 and Monoclonal Antibodies for Detecting Nipah and Hendra Viruses. Viruses 2024, 16, 794. https://doi.org/10.3390/v16050794), that is, NiV AgELISA can detect not only NiV. If they are not species specific, it can be assumed that they will be able to detect similar viruses.
Answer:
Yes, this is a very good question. In the context of our previously published NiV-Ag ELISA assay, mAb F20NiV-65 bound only to NiV but not to HeV when paired with Ephrin B2. The reason why F20NiV-65 showed different specificities in different assays may be due to the alteration or disruption of antibody binding under each assay condition. Factors such as pH, temperature, salt concentration, and the presence of detergents can all affect the strength and specificity of the antibody-antigen interaction. Specifically in this case, the altered binding profile of F20NiV-65 could be because of (1) the detergent used in the AgELISA, as detergents disrupt hydrophobic interactions and, in some cases, reduce the effectiveness of antibody binding; (2) the plastic wells used in the AgELISA, since binding of viruses to the well surface may change the conformation of the protein and affect antibody binding; or (3) conformational changes in F20NiV-65 following binding to colloidal gold particles, which can affect the strength and specificity of the interaction with the antigen. This explanation was included in the revised manuscript.
Question: Thus it may be necessary to check for cross-reactivity with other viruses besides Cedar virus (CedV), for example such as Parahenipaviruses Langya virus and Mòjiāng virus, which are found in China.
Answer:
Yes, we agree. Unfortunately, we were unable to test for cross-reactivity with other closely related henipaviruses, such as Langya virus (LayV) and Mojiang virus (MojV) because we do not have access to these viruses. While the inability of the ILF test to detect EBOV, FMDV, and AIV does not necessarily serve a differential diagnostic potential, it does further demonstrate the specificity of our ILF test for NiV and HeV. We will be looking for partners with relevant virus resources to help us validate the ILF test. We will be looking for partners with relevant virus resources to help us validate the ILF test.
Question: What is the rationale for choosing EBOV for specificity comparison? Or foot-and-mouth disease virus (FMDV) and avian influenza virus (AI)? Are they somehow included in the list of potential differential diagnostics? It would be good to explain the choice of these viruses
Answer:
While the inability of the ILF test to detect EBOV, FMDV, and AIV does not necessarily serve a differential diagnostic potential, it does further demonstrate the specificity of our ILF test for NiV and HeV. We will be looking for partners with relevant virus resources to help us validate the ILF test.
Reviewer 3 Report
Comments and Suggestions for Authors
This study aims to develop a point-of-care immunochromatographic lateral flow strip assay for detecting Nipah and Hendra viruses using monoclonal antibodies specific to these viruses. The study is straightforward, and the presentation is clear. The most critical aspect of this study is the sensitivity of the ILF test, which is compared to the antigen test rather than the gold standard PCR assays. The sensitivity of PCR assays must be significantly higher than that of antigen tests. Below are my additional comments:
- Where were the P4 viruses cultured, and where were the experiments conducted? Does this study require Institutional Review Board (IRB) approval?
- There is a typographical error in Line 125; it should read (F20Niv-65) instead of (F20NiV-650.
- References 18 and 23 are by the same authors and were published in the same year; therefore, the reference cited for this author in Table 1 is unclear.
- According to Table 1, F20NiV-65 does not react with HeV, as also indicated in reference 18. However, this study demonstrates a reaction against HeV. Please explain.
- The specificity test (Table 2) indicates that no false positives were detected in serum samples, likely because these samples are cleaner than other swabs (oral, nasal, rectal), which often harbor various organisms, particularly bacteria or other substances. In fact, 8.8% (10/114) of swab samples produced false positives (Table 2). Excluding serum samples, the specificity of swab samples is 92.0%. The specificity of other specimen types, such as cerebrospinal fluid (CSF), nasopharyngeal aspirates, or bronchial aspirates, remains unclear.
- Additionally, further studies on other viruses, such as respiratory viruses, should be conducted to verify the specificity of the assay.
Author Response
We would like to thank reviewers for taking the time to carefully read our manuscript. We have significantly revised the manuscript based on the comments of the reviewers. We appreciate the opportunity to resubmit the revised manuscript now.
- Question: Where were the P4 viruses cultured, and where were the experiments conducted? Does this study require Institutional Review Board (IRB) approval?
Answer:
All work with infectious risk group 4 viruses was performed in the containment level 4 (CL4) laboratories of the Canadian Science Centre for Human and Animal Health, Winnipeg, Manitoba, Canada, under the jurisdiction of PHAC or NCFAD. All work was performed according to institutionally approved standard operating protocols. This information has been included in the manuscript.
- Question: There is a typographical error in Line 125; it should read (F20Niv-65) instead of (F20NiV-650.
Answer:
The typo has been corrected.
- Question: References 18 and 23 are by the same authors and were published in the same year; therefore, the reference cited for this author in Table 1 is unclear.
Answer:
This issue has been addressed by adding reference numbers to Table 1.
- Question: According to Table 1, F20NiV-65 does not react with HeV, as also indicated in reference 18. However, this study demonstrates a reaction against HeV. Please explain.
Answer:
Yes, this is a very good question. In the context of our previously published NiV-Ag ELISA assay, mAb F20NiV-65 bound only to NiV but not to HeV when paired with Ephrin B2. The reason why F20NiV-65 showed different specificities in different assays may be due to the alteration or disruption of antibody binding under each assay condition. Factors such as pH, temperature, salt concentration, and the presence of detergents can all affect the strength and specificity of the antibody-antigen interaction. Specifically in this case, the altered binding profile of F20NiV-65 could be because of (1) the detergent used in the AgELISA, as detergents disrupt hydrophobic interactions and, in some cases, reduce the effectiveness of antibody binding; (2) the plastic wells used in the AgELISA, since binding of viruses to the well surface may change the conformation of the protein and affect antibody binding; or (3) conformational changes in F20NiV-65 following binding to colloidal gold particles, which can affect the strength and specificity of the interaction with the antigen. This explanation was included in the revised manuscript.
- Question: The specificity test (Table 2) indicates that no false positives were detected in serum samples, likely because these samples are cleaner than other swabs (oral, nasal, rectal), which often harbor various organisms, particularly bacteria or other substances. In fact, 8.8% (10/114) of swab samples produced false positives (Table 2). Excluding serum samples, the specificity of swab samples is 92.0%. The specificity of other specimen types, such as cerebrospinal fluid (CSF), nasopharyngeal aspirates, or bronchial aspirates, remains unclear.
Answer:
Yes, we agree. It is true. This information has been included in the manuscript (line 291-297, red).
- Question: Additionally, further studies on other viruses, such as respiratory viruses, should be conducted to verify the specificity of the assay.
Answer:
We agree, this is a very good point. Unfortunately, we were unable to test for cross-reactivity with other closely related henipaviruses, such as respiratory viruses, Langya virus (LayV) and Mojiang virus (MojV) because we do not have access to these viruses. We will be looking for partners with relevant virus resources to help us validate the ILF test.